# Investigation of the Genetic Architecture of Pigs Subjected to Breeding Intensification

**DOI:** 10.3390/genes13020197

**Published:** 2022-01-22

**Authors:** Anatoly Kolosov, Lyubov Getmantseva, Maria Kolosova, Timofey Romanets, Nekruz Bakoev, Elena Romanets, Ilona Bakoeva, Olga Kostyunina, Yuri Prytkov, Olga Tretiakova, Siroj Bakoev

**Affiliations:** 1Don State Agrarian University, 346493 Persianovski, Russia; kolosov777@gmail.com (A.K.); m.leonovaa@mail.ru (M.K.); timofey9258@mail.ru (T.R.); lena9258@mail.ru (E.R.); tretiakova.olga2013@yandex.ru (O.T.); siroj1@yandex.ru (S.B.); 2Centre for Strategic Planning and Management of Biomedical Health Risks Disabled, 119435 Moscow, Russia; 3Federal Research Center for Animal Husbandry Named after Academy Member LK. Ernst, 142132 Dubrovitsy, Russia; nekruz82@bk.ru (N.B.); kostolan@yandex.ru (O.K.); prytkov_y@mail.ru (Y.P.); 4Mendeleev University of Chemical Technology, 125480 Moscow, Russia; ilonaluba2@mail.ru

**Keywords:** pig, selection signals, inbreeding coefficient, homozygous-by-descent (HBD), smoothing FST

## Abstract

Pigs are strategically important animals for the agricultural industry. An assessment of genetic differentiation between pigs, undergone and not undergone to selection intensification, is of particular interest. Our research was conducted on two groups of Large White pigs grown on the same farm but in different years. A total of 165 samples were selected with 78 LW_А (*n* = 78, the Russian selection) and LW_B (*n* = 87, a commercial livestock). For genotyping, we used GeneSeek^®^ GGP Porcine HD Genomic Profiler v1 (Illumina Inc, San Diego, CA, USA). To define breeding characteristics of selection, we used smoothing FST and segment identification of HBD (Homozygous-by-Descent). The results of smoothing FST showed 20 areas of a genome with strong ejection regions of the genome located on all chromosomes except SSC2, SSC3, and SSC8. The average realized autozygosity in Large White pigs of native selection was in (LW_A)—0.21, in LW_В—0.29. LW_А showed 13,338 HBD segments, 171 per one animal, and LW_B—15,747 HBD segments, 181 per one animal. The ejections found by the smoothing FST method were partially localized in the HBD regions. In these areas, the genes ((NCBP1, PLPPR1, GRIN3A, NBEA, TRPC4, HS6ST3, NALCN, SMG6, TTC3, KCNJ6, IKZF2, OBSL1, CARD10, ETV6, VWF, CCND2, TSPAN9, CDH13, CEP128, SERPINA11, PIK3CG, COG5, BCAP29, SLC26A4) were defined. The revealed genes can be of special interest for further studying their influence on an organism of an animal since they can act as candidate genes for selection-significant traits.

## 1. Introduction

A human noticed long ago that some traits are displayed in animals of one species differently, and he selected the individuals with traits interesting to him and raised them artificially on his farm. Thus, there appeared a selection traditionally based on breeding and selection aimed at fixing desirable traits in the population. The development of genetics and molecular biology made it possible to considerably raise the efficiency of selection and breeding work, which also influenced the rate of improvement of animals. A long-term, targeted selection by the same breeding important traits leads to the appearance of the so-called selection signatures in the genome of farm animals and is associated with specific traits [1].

Selection signatures were found for the genes-candidates linked to such industrial characteristics as growth rate and development [2,3], reproduction [4], meat quality [5], biological regulation and metabolism [6], level of immunity, adaptability [7], processes providing development of a brain [8], and morphological variations, the form of ears [9], and body length [10].

When searching for selection signatures, the FST method is extremely interesting. Value FST is a differentiation measure between populations [11]. The FST locus value is calculated as the ratio of the variance of allele frequencies between populations and the sum of variances within and between populations. The locus with more significant values of FST in comparison with other loci may indicate positive selection [12,13]. To date, various modifications of this method have been presented, but undoubtedly, it remains the most widespread and reliable one to identify genome traits of selection signatures between observable populations [14,15].

Method FST is used to identify adjacent regions of the genome during selection and is useful for analyzing distantly related populations as it reveals subtle differences between them [16]. Smoothed method FST is based on the model of Nicholson pure drift [17], according to which separate SNP clusters in genome windows are calculated as average values.

Selection signatures can also be localized in homozygous areas of various lengths [1]. When creating breeds of farm animals, the accumulation of homozygosity allows pure-bred animals to possess certain qualities and steadily pass them on to their offspring [18]. The use of homozygosity patterns of ROH (Runs Of Homozygosity) makes it possible to reveal long homozygous areas in a genome [19]. In the research, it is reasonable to use the method proposed by Drouet and Gaultier based on the model of HBD multiple classes (homozygous-by-Descent). This method allows estimating autozygosity according to the age of the ancestors.

Pigs are strategically important animals for agriculture. In this connection, over the past decades, targeted work has been carried out to increase selection-significant indicators [20]. This enabled us to considerably increase the pigs’ reproductive, growth, and meat indexes. However, alongside high efficiency in pigs, there appeared various anomalies, congenital defects, problems with limbs, and susceptibility to various diseases [21]. Thereby an assessment of genetical differentiation between the pigs that have undergone and have not undergone selection intensification is of particular interest. In this aspect, we conducted research using pigs raised on the territory of the Russian Federation at different periods of time and identified the signatures of selection in pigs due to trends in different socio-economic conditions [22]. However, studying modern livestock of Large White pigs has shown that formations of selection signatures are influenced by selection centers themselves since each of them implements its own breeding strategy. In this connection, we focused our work on pigs raised on one farm of the Russian Federation, but at different periods of time. In addition to the FST method, we investigated homozygosity areas using a model of plural HBD classes, defined genome signatures identified by two methods, determined QTL enrichment, and positioned genes in these areas.

## 2. Materials and Methods

### 2.1. Animals

Anesthesia, euthanasia, or any animal sacrifice was not used to conduct this study. This study did not involve any endangered or protected species. According to standard monitoring procedures and guidelines, the participating holdings specialists collected tissue samples, following the ethical protocols outlined in the Directive 2010/63/EU (2010). The pig ear samples (ear pluck) were obtained as a general breeding monitoring procedure. The collection of ear samples is a standard practice in pig breeding [23].

### 2.2. Sampling and Genotyping

For our work, we chose Large White pigs, which were kept on the same farm but in different years. Pigs of the LW_A group belonged to the Russian selection, which is based on pigs of the Large White breed, imported from England in 1923–1931. Long-term breeding work, taking into account the local climate, changed the English type of Large White pigs, and, in fact, a new domestic Large White breed was created, which at that time surpassed the English in many respects. At the beginning of the 21st century, these pigs almost completely disappeared, and imported pigs began to be imported into the Russian Federation [24]. LW_A were distinguished by good adaptation and resistance to various diseases were less whimsical to the conditions of keeping and feeding, but the pigs of imported selection were significantly superior in growth rate, reproductive performance, and thinner fat. Group LW_A (date of assembly 2008–2010) and LW_B (date of assembly 2014–2016). Pigs of the LW_A group belonged to the Russian selection, and pigs LW_B belonged to commercial livestock, which was delivered to the farm from Europe in 2013. For work, 165 samples were selected, 78 LW_А and 87 LW_B. Genomic DNA was extracted from tissue (ear pinch) using a set of DNA-Extran-2 reagents (OOO NPF Sintol, Russia). For genotyping, we used GeneSeek^®^ GGP Porcine HD Genomic Profiler v1, which included 68,516 SNPs evenly distributed with an average spacing of 25 kb. (Illumina Inc, San Diego, CA, USA). The total genotyping rate was 0.99.

### 2.3. Data Analysis

To make relations between populations visual we conducted SVD (singular value decomposition) by means of basic svd function in R. The Heatmap graph was plotted on the basis of the GRM matrix. To define selection characteristics, we used methods of smoothing FST. For smoothing FST a filtration of the data hwe 1 × 10 ^−7^ maf 0.01—geno 0.2—mind 0.2—indep-pairwise 50 5 0.2 42,442 variants passed the QC filters and were retained for further analysis. To filter the noise obtained as a result of the FST calculation, the lokern smoothing algorithm was applied: Kernel Regression Smoothing with Local or Global Plug-in Bandwidth of the lokern package in R [25] with the n.out = 424 parameter, which approximately corresponds to one point for every 100 SNPs and allows smoothing SNP data set. The value of the x.out parameter was used to match the smoothed values against the SNP reference map and its position. The smoothest FST values, corresponding to 0.999%, were identified and translated into genomic positions of Sus scrofa 11.1, and the gene content of each region was analyzed.

To identify HBD segments and to assess autozygosity (or the coefficient of inbreeding), we used the multiple HBD classes model presented in RZooRoH package [26,27]. The method was insensitive to MAF filtration and rather resistant to the structure of LD. In this connection, the data filtration by MAF and LD was not conducted. The Rk coefficients were set from 2 to 516 (2, 4, 8, 16, 32, 64, 128, 256, 512). The inbreeding coefficient was calculated as the sum of autozygosity for all HBD classes. The total number of HBD segments, the average number per individual, the average length of the segment per individual, and the distribution of segments (and their average length) on the chromosomes were assessed for each group. We defined SNP frequency (%) in the found HBD segments and, for each group, chose top HBD provided that HBD frequency was not less than 60% and included at least 10 SNPs. Based on the results of the 2 methods (FST and HBD), regions of the genome and genes were identified, probably associated with the intensification of the selection process in commercial pigs.

### 2.4. Search and the Analysis of QTL Enrichment

The search of QTL, genes, and the QTL enrichment analysis performed in Genomic Annotation in Livestock for positional candidate LOci (GALLO) was an R package Ensembl genome browser [28], and also a literature search was also carried out manually for the presence of data on the associations of genes with any traits in humans and animals.

## 3. Results

To assess the genetic structure of the studied populations of Large White pigs, we used SVD and Heatmap. Figure 1A,B show that pigs of groups LW_A and LW_B have their own individual cluster.

The results of smoothing FST showed 20 regions of the genome with strong outliers located on all chromosomes, with the exception of SSC2, SSC3, and SSC8 (additional Appendix A). These areas overlap with quantitative trait loci (QTLs), of which Meat and Carcass traits were most represented (Figure 2A). Based on the analysis of QTL enrichment with the most significant enrichment, the signs of pH 24 hr post-mortem (lion), meat color a *, and Cortisol level were identified (Figure 2B).

On average, the realized autozygosity was 0.21 for LW_A and 0.29 for LW_B (Figure 3). The maximum values (0.34) were recorded in the LW_B group and the minimum (0.15) in the LW_A group.

The class Rk = 128 contributed greatly to autozygosity in LW_B (proportion of the genome about 0.1) (Figure 4A,B; additional Appendix A). Herewith, the contribution of the classes Rk = 128 (0.05) and the class Rk = 256 (0.05) can be traced in LW_A. Variations in individual levels of autozygosity in both groups did not deviate much from the midline (Figure 4C,D). HBD class Rk = 512 was absent in all pigs in the study groups.

In general, LW_A had 13,338 HBD segments, an average of 171 per animal; LW_B has 15,747 HBD segments, an average of 181 per animal. The largest length of HBD segments was determined for LW_A (138.52 Mb, 1759 Number SNP, SSC1). The average length of HBD segments for LW_A was about 2.47 Mb (54 Number SNP), for LW_B 3.41 Mb (77 Number SNP). The average length of HDB segments on chromosomes (taking into account different classes) is shown in Figure 5. Segments of the Rk = 2 class were determined only in pigs from the LW_A group. These segments were located in SSC1, and their average length was 108.67 Mb. Segments of class Rk = 4 for LW_A were defined at SSC1, SSC2, SSC4, SSC5, SSC6, SSC9, SSC13, SSC17; LW_B has SSC1, SSC2, SSC8, SSC9, SSC13, and SSC15. HBD segments of class Rk = 8 were absent in LW_A on SSC10 and in LW_B on SSC10 and SSC16. Starting from Rk = 16 and further up to Rk = 256, HBD segments were relatively evenly distributed on all chromosomes in LW_A and LW_B.

The SNP frequencies (%) in the detected HBD were estimated for each group of pigs and plotted against the position of the SNP in the autosomes (Figure 6 and Figure 7).

For each group, the top HBD were selected, provided that the HBD frequency was at least 60% and at least 10 SNPs were included. As a result, LW_A has 4 regions located in SSC1 (Table 1). LW_B has 10 regions, of which 5 were in SSC1, 2 in SSC6, and one each in SSC10, SSC14, and SSC15. The topHBD regions did not overlap between groups.

In both groups, the top HBD areas overlapped with QTLs, among which the most represented were the signs of Meat and Carcass traits (Figure 8). In pigs LW_A, relative to LW_B, the QTL type of Exterior, Health and Production was more represented. In LW_B pigs, QTL Reproduction was more represented. Using the analysis of QTL enrichment in the top HBD, LW_A had the most represented characteristics: Ph Logissmus Dorsi, carcass weight (hot), body weight (weaning), backfat at last rib, average daily gain. LW_B has shoulder subcutaneous fat thickness, shear force, loin muscle area, fat area percentage in the carcass, estimated carcass lean content, and dressing percentage.

In the top HBD, genes encoding proteins and having names (according to Ensembl genome browser 104) were identified, but genes encoding lncRNA, snoRNA, snRNA were also represented in these regions with a high frequency (additional Appendix A).

The ejections detected by the smoothing FST method were partially localized in the HBD regions. Table 2 shows the areas in which high outliers were found in FST smoothing and their frequencies in the HBD areas.

## 4. Discussion

The average length of HBD, chromosome distribution, and the genome proportion covered by HBD can be used as indicators of the origin and history of a population, as well as reflecting events of artificial selection. The Large White breed was created in the 1870s–1880s and officially received its name in 1885 [24]. According to the results of our research, the shortest HBD was inherited from ancestors about 128 years ago, which in general was exactly the period of the formation of the Large White breed. Subsequently, the Large White breed took part in the creation and improvement of most modern European breeds, as well as local breeds created on the territory of the USSR. Our studies showed that the total autozygosity of LW_B pigs was 0.29 proportion of the genome, while the contribution of the Rk = 128 class segments was about 0.1. On this basis, we may assume that ancestors contributed greatly to the autozygosity of this group about 64 years ago. This period was superimposed on the period of growth of intensification processes in pig breeding and the formation of commercial livestock, characterized by high production indicators [29]. In its turn, the total autozygosity in pigs of the LW_A group was 0.21 proportion of the genome, but the dominant contribution of any of the classes was not observed because the contribution of ancestors about 64 (Rk = 128) and 128 (Rk = 256) years ago amounted to 0.05 shares of autozygosity. It can be assumed that the period of intensification was reflected in the LW_A population, but in a much smaller volume, which made it possible to preserve the signatures inherited from more distant ancestors.

Common autozygosity areas in a population identify selection hotspots [30]. In the groups under study, the top HBD was determined, provided that their frequency was less than 60% and included at least 10 SNPs. In general, in both groups, QTLs for Meat and Carcass were most represented, but in group LW_A they stood out to a greater extent with traits associated with carcass weight, and in LW_B—traits associated with obtaining lean pork.

Ejections identified by FST smoothing were partially localized in HBD segments. For example, SNP rs81349176 (SSC1) occured with a frequency of 0.73 in HBD in LW_A. The functional significance of SNP rs81349176 was difficult to interpret since it was localized in the intragenic region. However, it is interesting to mention that the adjacent genes CDH19 and CDH7 belong to the cadherin family, which plays a key role in the regulation of adhesion. Dysregulation of adhesion molecules often causes various diseases, including inflammation and tumors [31,32,33]. Earlier, we also suggested a hypothesis about the connection of cadregins with Capped Hock in Pig [34]. In the future, it is vital to study cadregins in more detail in terms of limb tumors since this is a source of significant economic losses in pig production [35,36].

Genes were determined in the regions identified by smoothing FST (NCBP1, PLPPR1, GRIN3A, NBEA, TRPC4, HS6ST3, NALCN, SMG6, TTC3, KCNJ6, IKZF2, OBSL1, CARD10, ETV6, VWF, CCND2, TSPAN9128, CDH13, PIK3CG, COG5, BCAP29, SLC26A4), which may be selection signals related to growth, conformation, health, reproductive performance, and meat quality. The identified genes were associated with the structural and functional work of the cell (CARD10, TSPAN9, PIK3CG, COG5, BCAP29). These genes are involved in the transmission of apoptosis signals (CARD10, BCAP29), play a certain role in the regulation of development, activation, growth, and motility of cells (TSPAN9), as well as in maintaining the structural and functional integrity of the epithelium and regulation of cytotoxicity in NK cells (PIK3CG), morphology and functions of the Golgi complex (COG5) [37].

It is interesting to note the variants of the genes influencing the function of a hemopoiesis and hereditary diseases of the circulatory, cardiovascular system, and other pathologies (HS6ST3, IKZF2, ETV6, SMG6, SLC26A4, VWF). The genes participate in proliferation and differentiation, adhesion, migration, inflammation, fibrillation, and other various processes (HS6ST3), in the regulation of development of lymphocytes (IKZF2), play a role in the hemopoiesis and malignant transformation (ETV6), and are associated with an increased risk of ischemic heart disease (SMG6), inherited hearing loss in domestic animals (SLC26A4).

The genes regulating important components of the nervous system have also been identified (KCNJ6, TRPC4, NALCN, GRIN3A, NBEA). The gene mutations are associated with severe developmental delay, facial dysmorphism and mental retardation, reduced cognitive ability (KCNJ6, NALCN). They play an important role in dopamine-related processes, including addiction and attention (TRPC4), physiological and pathological processes in the central nervous system (GRIN3A), autism (NBEA), etc.

The revealed genes can be of special interest for further studying of their impact on an organism of an animal as they can represent themselves as genes-candidates bound to the physiological features of an organism providing high industrial indexes, but also probably associated with changes of the nervous system, various pathologies, including the number of circulatory and cardiovascular systems.

## 5. Conclusions

Investigations aimed at studying the presence and localization of selection signatures (FST), as well as the identification of areas of homozygosity (HBD) in two groups of pigs bred at different times on the same farm, enabled us to identify differences between populations. The presence of QTLs located in areas of homozygosity and associated with traits, the improvement of which was aimed at selection breeding work, makes such areas the most promising for the search for potential candidate genes associated with the level of productivity and the presence of diseases.

In the genome regions determined by using the FST and HBD methods, we identified genes that may have contributed to the changes associated with the intensification of the selection process in pigs. In general, the results presented in our work show promising prospects for genome scanning using FST and HBD methods for studying population history, as well as for identifying genomic regions and genes associated with important economic traits and various pathologies.

## Figures and Tables

**Figure 1 genes-13-00197-f001:**
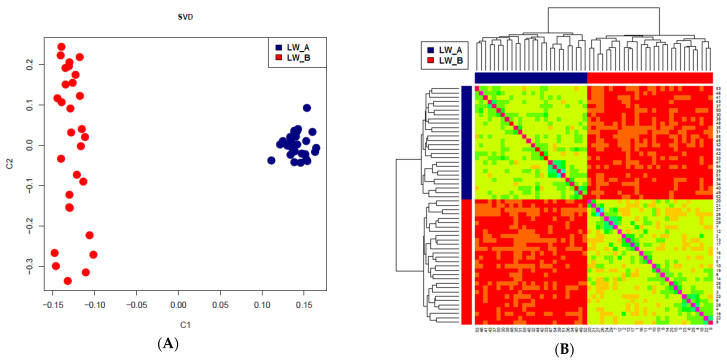
SVD (**A**) and Heatmap (**B**) for pigs LW_A and LW_B. (LW_A—Large White Russian selection, LW-B—a commercial Large White).

**Figure 2 genes-13-00197-f002:**
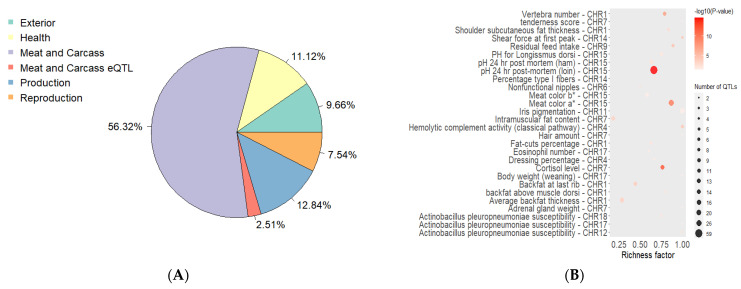
QTLs in genomic regions with strong outliers according to smoothing FST. *(*(**A**)—percent QTL type. (**B**)—QTL enrichment analysis (the more intensive the red shade, the more significant is enrichment; the area of circles is proportional to quantity QTL; richness factor—the attitude of QTL quantity, annotated in the study areas, to the total number of each QTL in the reference database).

**Figure 3 genes-13-00197-f003:**
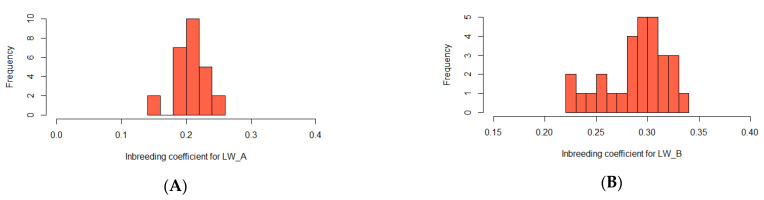
Implemented autozygosity in pigs. *(*(**A**)—distribution of the values of the inbreeding coefficient for LW_A; (**B**)—distribution of the values of the inbreeding coefficient for LW_B).

**Figure 4 genes-13-00197-f004:**
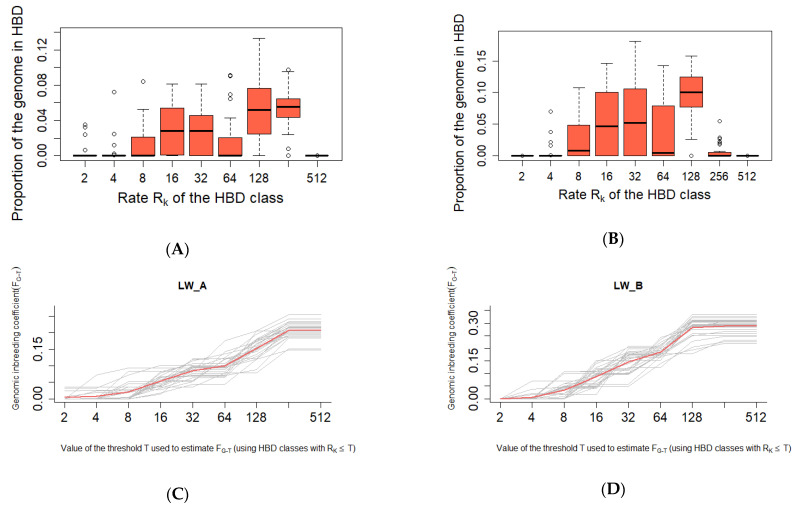
The proportion of autozygosity in different HBD classes. ((**A**)—the proportion of autozygosity in different classes of HBD in LW_A; (**B**)—the proportion of autozygosity in different classes of HBD in LW_B; (**C**)—variations in individual levels of autozygosity in LW_A; (**D**)—variations in individual levels of autozygosity in LW_B).

**Figure 5 genes-13-00197-f005:**
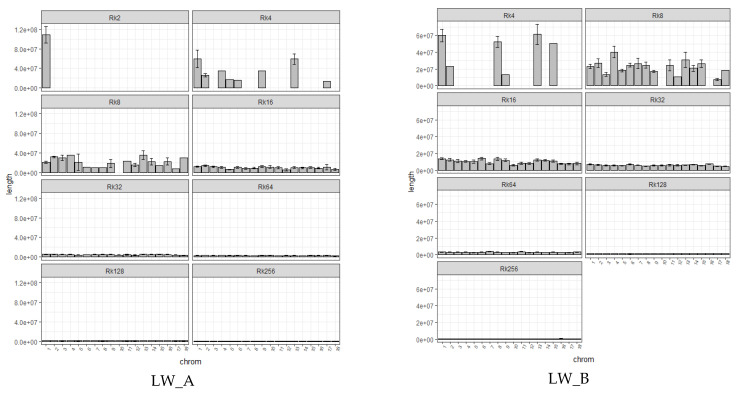
The average length of HDB segments on chromosomes in different classes in pigs.

**Figure 6 genes-13-00197-f006:**
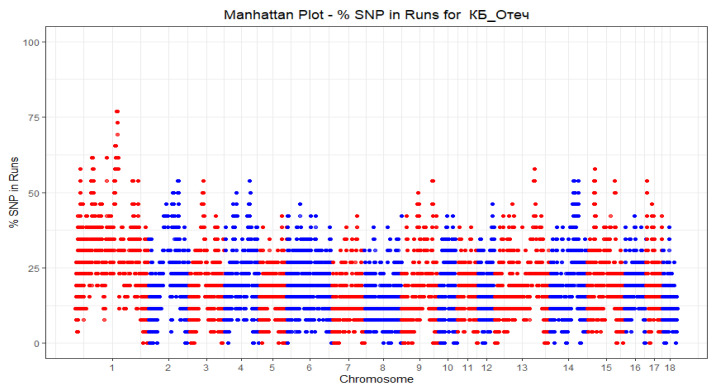
Manhattan Plot-% SNP in HBD for LW_A.

**Figure 7 genes-13-00197-f007:**
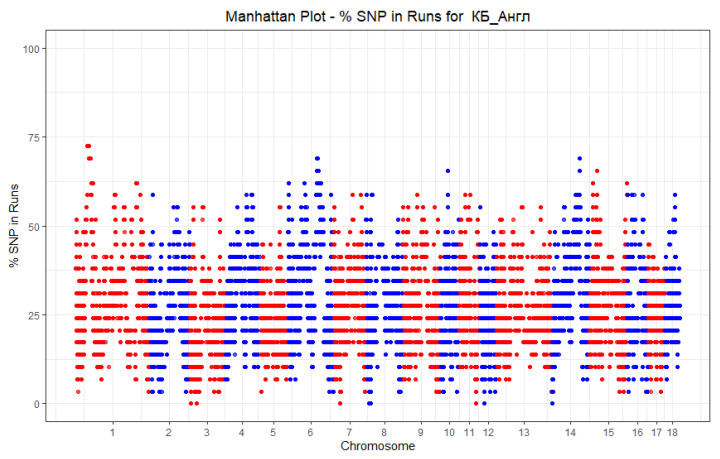
Manhattan Plot-% SNP in HBD for LW_B.

**Figure 8 genes-13-00197-f008:**
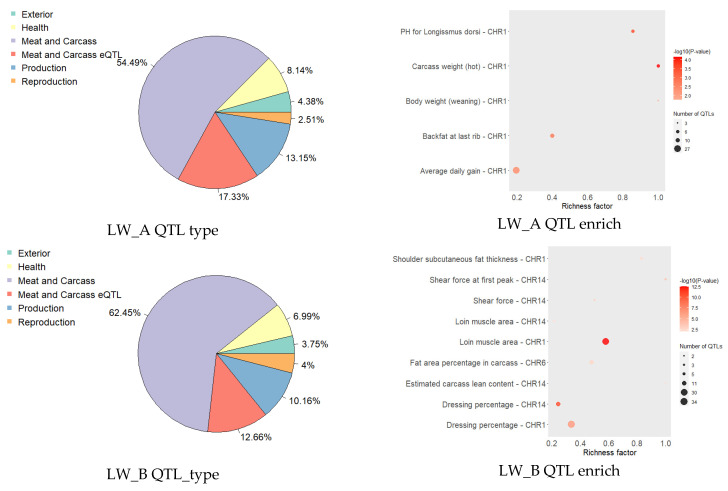
QTLs type in HBD areas.

**Table 1 genes-13-00197-t001:** Top HBD pig LW_A and LW_B.

Group	Chrom	nSNP	From	To	Gene
LW_A	1	13	63704733	64301394	*FHL5*, *GPR63*, *NDUFAF4*, *KLHL32*
LW_A	1	14	116300444	117138985	*PYGO1*, * DNAAF4*, * C15orf65*, * CCPG1*, * PIGB*, * RAB27*, * RSL24D1*
LW_A	1	17	149261646	150160033	*ZNF407*, * CNDP1*, * CNDP2*, * DIPK1C*, * C18orf63*, * CYB5A*, * FBX015*, * TIMM21*
LW_A	1	46	154796451	158940596	*DSEL*, * CDH19*, * CDH7*, * SERPINB10*, * SERPINB7*, * SERPINB12*, * SERPINB8*, * SERPINB5*, * SERPINB2*, * SERPINB13*, * SERPINB11*, * VDS4B*, * PHLPP1*, * KDSR*
LW_B	1	245	41940391	53513159	*MAN1A1*, * FAM184A*, * ASF1A*, * CEP85L*, * SLC35F1*, * NUS1*, * DCBLD1*, * ROS1*, * VGLL2*, * RFX6*, * GPRC6A*, * FAM162B*, * KPNA5*, * ZUP1*, * RSPH4A*, * PTP4A1*, * PHF3*, * ADGRB3*, * LMBRD1*, * COL19A1*, * COL9A1*, * FAM135A*, * SMAP1*, * B3GAT2*, * OGFRL1*, * RIMS1*, * KCNQ5*, * DPPA5*, * OOEP*, * CYB5R4*, * MRAP2*, * CEP162*
LW_B	1	10	56563956	57069421	*RNGTT*
LW_B	1	11	61986232	62371709	-
LW_B	1	14	225017887	225614369	*CEMIP2*, * ABHD17B*, * C9orf85*, * GDA*
LW_B	1	18	227781084	229301982	*TRPM6*, * C9orf40*, * NMRK1*, * CARNMT1*, * OSTF1*, * PCSK5 *
LW_B	6	20	61261671	63135016	*PEG3*, * AURKC*, * ZNF304*, * ZNF772*, * ZNF773*, * ZNF550*, * ZNF606*, * ZNF135*, * ZNF329*, * ZNF274*, * ZNF8*, * RPS5*, * ZNF584*, * ZNF446*, * SLC27A5*, * ZBTB45*, * TRIM28*, * CHMP2A*, * UBE2M*, * MZF1*
LW_B	6	84	107551091	113309905	*RBBP8*, * CABLES1*, * TMEM241*, * RIOK3*, * RMC1*, * NPC1*, * ANKRD29*, * LAMA3*, * TTC39C*, * CABYR*, * OSBPL1A*, * HRH4*, * ZNF521*, * SS18*, * PSMA8*, * TAF4B*, * KCTD1*, * AQP4*, * CHST9*, * CDH2*
LW_B	10	14	28373981	28880540	-
LW_B	14	19	104827372	105533891	*MYOF*, * CEP55*, * FFAR4*, * RBP4*, * PDE6C*, * FRA10AC1*, * LGI1*, * SLC35G1*, * PLCE1*
LW_B	15	22	26069754	26699994	-

**Table 2 genes-13-00197-t002:** Intersection of the smoothing FST and HBD areas.

		LW_A	LW_B	Gene
1	SSC1: 239389749–243367727	0.38	0.51	*NCBP1*, * PLPPR1*, * GRIN3A*
2	SSC1: 155986286	0.73	0.31	-
3	SSC4: 7921863	0.08	0.07	-
4	SSC4:40772747–41434985	0.27	0.45	-
5	SSC4: 119338511	0.23	0.31	*lncRNA*
6	SSC5: 60515812–67077093	0.19	0.48	*ETV6*, * VWF*, * CCND2*, * TSPAN9*
7	SSC5: 10294138	0.12	0.31	*CARD10*
8	SSC6: 5154694	0.12	0.38	*CDH13*
9	SSC7: 103380850–115666493	0.24	0.07	*CEP128*, * SERPINA11*
10	SSC9: 106642564–119757434	0.08	0.52	*PIK3CG*, * COG5*, * BCAP29*, * SLC26A4*
11	SSC10: 49802418–50277330	0.23	0.31	-
12	SSC11: 11335614	0.23	0.17	*NBEA*
	SSC11: 13345472–13527422	0.23	0.31	*TRPC4*
	SSC11: 65881573–69872868	0.30	0.66	*HS6ST3*, * NALCN*
13	SSC12: 48181049	0.12	0.38	*SMG6*
14	SSC13: 200778735–201245827	0.04	0.48	*TTC3*, * KCNJ6*
15	SSC14: 13891794	0.12	0.21	-
16	SSC15: 115593713–121561503	0.23	0.38	*IKZF2*, * OBSL1*
17	SSC17: 8162613	0.19	0.31	-
18	SSC18: 45801997	0.08	0.24	-

## Data Availability

The raw data supporting the conclusions of this article will be made available by the authors upon reasonable request.

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
