# Peer review of "Investigation of the Genetic Architecture of Pigs Subjected to Breeding Intensification"

_genes, 2022, doi:10.3390/genes13020197_

Round 1
Reviewer 1 Report
This is a very interesting study to discover the effects of modern breeding selection on the genetic architecture of Large White pigs. The authors identified 20 areas of a genome with strong ejection regions of the Sus scrofa genome. The ejections found by the smoothing FST method are partially localized in the HBD regions. In these areas the several genes have been defined. Overall, this study is intriguing.
However, there are two points needed to specify: first, were there genetic links between the Large white herds of Russia and the Large white herds of Europe. Were they from the same genetic resource? Second, how different are the production performances of the two pig groups? This should be stated briefly in the introduction or result.
Meterials and Methods: it is unclear to me how to control the quality of SNP chip data, or what is the standard of quality control? Details about the data of SNP chip analysis need to be added.
3.1 Data analysis and 3.2 Search and the analysis of QTL enrichment
These should go in the Materials and Methods section.
One of the important purposes of selection signal detection is to identify the genomic imprints left by selection on a population genome, which reflects the genetic polymorphism at the population level. Since there were two pig groups in this study, the authors should explain the genetic relationship between the two pigs groups.
How do you deal with the outliers of FST values at each SNP site?
It would be good to have more detailed information about how to locate the selection signal region screened by smoothing FST to the core SNPs, and with the location of the core SNP as the center, how many fragments upstream and downstream are used to locate candidate genes?
Because the sex chromosome involves some genes of important economic traits, it is necessary to detect the X chromosome selection signal of female livestock population and increase the genes related to reproductive traits.
Author Response
Reviewer 1.
Thank you for your attention to our work.
1) However, there are two points needed to specify: first, were there genetic links between the Large white herds of Russia and the Large white herds of Europe. Were they from the same genetic resource? Second, how different are the production performances of the two pig groups? This should be stated briefly in the introduction or result.
Added
Pigs of the LW_A group belonged to the Russian selection, which is based on pigs of the Large White breed, imported from England in 1923-1931. Long-term breeding work, taking into account the local climate, changed the English type of large white pigs and, in fact, a new domestic large white breed was created, which at that time surpassed the English in many respects. At the beginning of the 21st century, these pigs almost completely disappeared, and imported pigs began to be imported into the Russian Federation. LW_A were distinguished by good adaptation and resistance to various diseases, were less whimsical to the conditions of keeping and feeding, but the pigs of imported selection were significantly superior in growth rate, reproductive performance and thinner fat.
2) Meterials and Methods: it is unclear to me how to control the quality of SNP chip data, or what is the standard of quality control? Details about the data of SNP chip analysis need to be added.
Added
For genotyping, we used GeneSeek® GGP Porcine HD Genomic Profiler v1, which in-cludes 68,516 SNPs evenly distributed with an average spacing of 25 kb. (Illumina Inc, the USA). The total genotyping rate is 0.99.
For smoothing FST a filtration of the data hwe 1e-07 - maf 0.01 - geno 0.2 - mind 0.2 - in-dep-pairwise 50 5 0.5; 42,442 variants passed the QC filters and were retained for further analysis.
3) 3.1 Data analysis and 3.2 Search and the analysis of QTL enrichment. These should go in the Materials and Methods section.
Сhanged
Materials and Methods
2.3 Data analysis and 2.4 Search and the analysis of QTL enrichment.
4) One of the important purposes of selection signal detection is to identify the genomic imprints left by selection on a population genome, which reflects the genetic polymorphism at the population level. Since there were two pig groups in this study, the authors should explain the genetic relationship between the two pigs groups.
To assess the genetic structure of the studied populations of Large White pigs we used SVD and Heatmap.
5) How do you deal with the outliers of FST values at each SNP site?
Added to Materials and Methods
To filter the noise obtained as a result of the FST calculation, the lokern smoothing algorithm was applied: Kernel Regression Smoothing with Local or Global Plug-in Bandwidth of the lokern package in R [] with the n.out=424 parameter, which approximately corresponds to one point for every 100 SNPs and allows smoothing SNP data set. The value of the x.out parameter is used to match the smoothed values against the SNP reference map and its position.
6) It would be good to have more detailed information about how to locate the selection signal region screened by smoothing FST to the core SNPs, and with the location of the core SNP as the center, how many fragments upstream and downstream are used to locate candidate genes?
The smoothest FST values, corresponding to 0.999% were identified and translated into genomic positions of Sus scrofa 11.1, and the gene content of each region was analyzed. In the FST and HBD smoothing regions, we determined the genes using the functions of the GALLO program (-find_genes_qtls_around_markers) and interval = 5000 db.
7) Because the sex chromosome involves some genes of important economic traits, it is necessary to detect the X chromosome selection signal of female livestock population and increase the genes related to reproductive traits.
In this study, we selected only autosomal chromosomes. In further studies, we will take into account the use of the X chromosome.
Reviewer 2 Report
This research is important when it comes to pig genomics. Here are some comments that could help improve the manuscript.
Do not start a sentence with a number or abbreviation. Please rephrase that.
Please, delete first two sentences in Material and Methods (“Anesthesia, euthanasia, or any animal sacrifice was not used to conduct this study. This study does not involve any endangered or protected species.”). We need to state what we used in the study, not what we didn’t.
The section Material and Methods must be improved. What were the criteria for selecting animals (age, whether they were related ...)?
The process of DNA sampling and extraction itself must be explained in more detail.Quality control?
Subchapters 3.1 and 3.2 should be found in the Material and Methods section.
Author Response
Reviewer 2.
Thank you for your attention to our work.
1) Do not start a sentence with a number or abbreviation. Please rephrase that.
Сhanged
2) Please, delete first two sentences in Material and Methods (“Anesthesia, euthanasia, or any animal sacrifice was not used to conduct this study. This study does not involve any endangered or protected species.”). We need to state what we used in the study, not what we didn’t.
It is a mandatory requirement for all animal research that all submitted papers comply with the ethical protocols set out in Directive 2010/63/EC (2010).
3) The section Material and Methods must be improved. What were the criteria for selecting animals (age, whether they were related ...)?
Added to Material and Methods
For our work we chose Large White pigs which were kept in the same farm, but in different years. Pigs of the LW_A group belonged to the Russian selection, which is based on pigs of the Large White breed, imported from England in 1923-1931. Long-term breed-ing work, taking into account the local climate, changed the English type of Large White pigs and, in fact, a new domestic Large White breed was created, which at that time sur-passed the English in many respects. At the beginning of the 21st century, these pigs al-most completely disappeared, and imported pigs began to be imported into the Russian Federation [27]. LW_A were distinguished by good adaptation and resistance to various diseases, were less whimsical to the conditions of keeping and feeding, but the pigs of imported selection were significantly superior in growth rate, reproductive performance and thinner fat.
4) The process of DNA sampling and extraction itself must be explained in more detail. Quality control?
Added
Genomic DNA was extracted from tissue (ear pinch) using a set of DNA-Extran-2 reagents (OOO NPF Sintol, Russia). For genotyping, we used GeneSeek® GGP Porcine HD Ge-nomic Profiler v1, which includes 68,516 SNPs evenly distributed with an average spacing of 25 kb. (Illumina Inc, the USA). The total genotyping rate is 0.99.
For smoothing FST a filtration of the data hwe 1e-07 - maf 0.01 - geno 0.2 - mind 0.2 - in-dep-pairwise 50 5 0.5; 42,442 variants passed the QC filters and were retained for further analysis.
5) Subchapters 3.1 and 3.2 should be found in the Material and Methods section.
Сhanged
Materials and Methods
2.3 Data analysis and 2.4 Search and the analysis of QTL enrichment.
Round 2
Reviewer 1 Report
Thanks for the authors' reply. After the revision, the problems I raised have been solved.